# Perspectives and Factors Affecting the Preventive Behavior Pertinent to COVID-19 among School Employees in Chiang Mai, Thailand: A Cross-Sectional Study

**DOI:** 10.3390/ijerph19095662

**Published:** 2022-05-06

**Authors:** Pheerasak Assavanopakun, Tharntip Promkutkao, Suchat Promkutkeo, Wachiranun Sirikul

**Affiliations:** 1Department of Community Medicine, Faculty of Medicine, Chiang Mai University, Chiang Mai 50200, Thailand; pheerasak.assava@cmu.ac.th (P.A.); tharntip.p@cmu.ac.th (T.P.); 2School of Economics, Chiang Mai University, Chiang Mai 50200, Thailand; suchat.promkutkeo@cmu.ac.th; 3Center of Data Analytics and Knowledge Synthesis for Health Care, Chiang Mai University, Chiang Mai 50200, Thailand

**Keywords:** school employees, COVID-19, preventive behaviors, school re-opening

## Abstract

The school is one of the most challenging environments for management to prevent the spread of COVID-19 infection. School employees play important roles as the main practitioners of recommended preventive measures. Consistent application of preventive measures by school employees strengthens the school’s readiness to reopen for on-site education. The study objectives are (1) to assess school employee agreement and actions in accordance with the preventive measures and recommendations for COVID-19 prevention and control, (2) to assess the readiness of the school and employees for on-site education, and (3) to determine factors associated with consistent performance of these measures. A cross-sectional study was conducted via an online survey from 5 November 2021 to 25 January 2022. Self-reported online questionnaires were used to collect school employees’ data. The readiness of schools for on-site education was assessed using 44 indicators from the practical guidelines by the Thai Department of Health. Of the 402 study participants, the majority of participants had agreed to all measures and recommendations for COVID-19 prevention and control in school. High levels of concern and disagreement in school re-opening (aOR 3.78, 95%CI 1.04 to 13.70; *p* = 0.043) were associated with higher consistent performance of the measures and recommendations for COVID-19 prevention and control in schools. Male teachers (aOR 0.43, 95%CI 0.23 to 0.80; *p* = 0.008) and any disagreement with these measures and recommendations (aOR 0.03, 95%CI 0.01 to 0.23; *p* < 0.001) were associated with lower consistent performance. Our study findings can inform the stakeholders to decide on the re-opening and supportive strategies. School employees, especially in male employees, must be supported from the relevant sectors to decrease disagreement to the measures and enhance awareness of the pandemic situation before the school re-opening.

## 1. Introduction

Since the COVID-19 virus was first recognized in Wuhan city, the pandemic was started spreading from China to other countries including Thailand. The number of people infected and also the number of dead have been astronomical, fluctuating between 2020 when the pandemic first began until now. Many preventive measures were instigated to control the pandemic together with screening for the disease and immunization. The Delta strain of COVID-19 in Thailand (September 2021) had been transmitted and infected approximately 1,500,000 people the number of dead totalling around 16,000 [1]. During the same period in Chiang Mai, a total of 8300 were infected with the number of dead totalling 40 [2]. These included those within the high-risk group of infection such as elderly and children as dependents which experienced difficulties in management after infection. Students were also classed as a high-risk group due to range of age which had limitations as regards immunization and could be responsible for connections transferring infection to others [3]. Due to the pandemic, academic patterns were affected and there was a need to transform from on-site to online patterns of learning. The normal personal lifestyle was also changed to ensure consistency with the general measures issued for the prevention of the spread of the disease including wearing a mask, frequently washing hands, and maintaining distance between other people. It is a complicated issue preventing transmission from adult to children or people under supervision and also regarding vaccine issues such as vaccine coverage or types of vaccine for children under 18 years old. When considering the return to on-site education, preparation to prevent disease transmission by all stakeholders needs to be precise before making decisions regarding school re-opening.

Transformation of educational pattern during pandemic had many advantages such as giving a high level of flexibility to the learners, easy access for the learners, learner-centered, etc., but the limitations still exist including less response from the learners to the teacher, lack of attentiveness or discipline, and more expense for gadgets to use in online learning [4]. On-site teaching is still an important way for children to learn. Management in formatting the pattern of education must be adapted with the pandemic situation within the specific area which must concur with standard recommendations or practices for employees in the education context. An example is the standard recommendations or practices including guidance for COVID-19 prevention in K-12 schools by Centers for Disease Control and Prevention (CDC) [5] which stated that the main consideration was the safety of student during on-site education followed by general preventive measures (wearing a mask, social distancing, practices when employees had abnormal symptoms, and preparation of the site including a proper ventilation system and screening for diseases). Occupational Safety and Health Administration (OSHA), the agency that oversee labor in the United States of America also recommend the following of CDC as appropriate practices [6]. In Thailand, the Ministry of Public Health in the Department of Health announced practices of readiness for on-site education for either school or teacher [7] which detailed preventive measures which needed to be consistently performed to prevent the spread of the disease. The perspective and compliance when it came to individual preventive behavior can potentially inform how the community would cope with the pandemic situation [8,9,10]. Both the general population [11] and workers [12] demonstrated low compliance with preventive measures. There are various factors associated with preventive measures compliance, including external factors such as incomes, educational level, occupation, or internal factors including gender, age range, and individual knowledge about disease. These factors should be considered to develop supportive strategies for practitioners [13].

Due to a lack of information regarding agreement and compliance with preventive measures among key players, including school employees and schools, estimating the level of efficacy of the preventive measures and the risk of an outbreak following the school’s reopening were challenging. This study aimed (1) to assess school employee agreement and actions in accordance with the preventive measures and recommendations for COVID-19 prevention and control, (2) to assess the readiness of the school and employees for on-site education, and (3) to determine factors associated with consistent performance of these measures. The overall situation during the COVID-19 pandemic is very changeable due to the rapid adaptation of SARS-CoV-2 variants [14]. The percentages of preventive measures performed by school employees and the level of school readiness could reflect the efficacy of the current preventive measures for school re-opening during the pandemic of COVID-19. Identifying factors related to school employees’ compliance with preventive measures is also essential for enhancing their compliance. Our study findings can inform the stakeholders to decide on the re-opening and develop supportive strategies.

## 2. Materials and Methods

### 2.1. Study Design and Participants

This cross-sectional study was conducted via an online survey from 5 November 2021 to 25 January 2022, after the peak period of the 4th wave of COVID-19 (Delta variant) and early on in the 5th wave (Omicron) of the outbreak in Thailand [15]. An online survey was distributed to the school employees who had a responsibility for COVID-19 prevention in each school. Survey data were collected and managed using REDCap (Research Electronic Data Capture), which is a secure, web-based software platform designed to support data capture for research studies. This study invited 5767 school employees in all 72 schools located in urban district of Chiang Mai. They were informed to participate this online surveys via the school coordinators if they had a responsibility for COVID-19 prevention in their schools. Of 456 responses, 431 participants from 53 schools gave their consent to participate in this study. Four hundred and two (93.3%) participants completed all the questions. Figure 1 shows timeline of data collection and the periods of the COVID-19 outbreak in Thailand. Apart from that, information from school section was collected by a self-report questionnaire which was distributed to all 72 schools located in urban district of Chiang Mai and assigned to be answered by who had a responsibility for COVID-19 prevention in each school.

### 2.2. Questionnaire Design

The participants were asked about their behavior in line with the general preventive measure and their opinion regarding the preventive measures announced by the health sector. The questionnaire was made up of three main parts:(1)Personal information including socio-demographic characteristics (age, gender, and household income), occupational information (length of working’s experience, school affiliation, and school roles), personal preventive activities (COVID-19 vaccination status and cost of COVID-19 prevention per month)(2)School employees perceptions of the readiness of the school and employees for the re-opening of the school for on-site education (possible answers were the four categories: “strongly agree”, “agree”, “disagree”, and “strongly disagree”) and the employees concerns regarding the re-opening of schools for on-site education (again, possible answers were from four categories: “very concerned and do not agree with school opening”, “very concerned but agree with school opening”, “moderately concerned and agree with school opening”, and “slightly concerned”)(3)The questions for assessing school employees’ agreement and actions on COVID-19 preventive practices were designed in accordance with the measures and recommendations for school employees in the prevention of the spread of the epidemic of COVID-19 in Thailand. These measures and recommendations were developed by the national committee, including public health experts and infectious disease specialists from the department of health, and educational experts from the ministry of education. These consisted of eleven preventive measures to be performed by the employees to prevent the spread of the disease in school. This part of questionnaire was divided into two sub sections including the participant’s opinion with regard to every measure (possible answers being “agree” or “disagree” to each measure) and what were the actions of the participants as regards the carrying out of these preventive measures (possible answers were “consistently performed”, “partially performed”, and “not performed”). The reliability test of 11 questions for assessing school employees’ agreements and actions in accordance with the preventive measures obtained a Cronbach’s coefficient alpha of 0.83 and 0.89, respectively (Appendix A).

For the purpose of school assessment, the self-report questionnaires were used to obtain school preventive preparation data according to the practical guidelines by the department of health of Thailand. These measures and recommendations were developed by the national committee, including public health experts and infectious disease specialists from the department of health, and educational experts from the ministry of education. The questionnaire had 44 questions and consisted of one primary dimension (safety dimension) and five secondary dimensions (learning, coverage for the underprivileged, welfare, policies, and financial management). The reliability test of the six dimensions of school assessment obtained a Cronbach’s coefficient alpha between 0.76 and 0.82, respectively (Appendix A). If the school can follow all dimensions, the green criterion was labeled. If the school can follow the primary dimension but not all secondary dimensions, the yellow criterion was labeled. If the school cannot follow the primary dimension, it was labeled as the red criterion. The detail of “Self-assessment form for educational institutions to prepare before re-opening to monitor and prevent COVID-19 spreading 2020” is provided in the Appendix A).

### 2.3. Statistical Analysis

All statistical analyses were conducted using the STATA statistical software program (Stata Corp. 2019, Stata Statistical Software: Release 16, Stata Corp LLC, College Station, TX, USA). The personal information, perceptions by the school employees, opinions, and actions concordant with the practice by the school employees in prevention of the spread of the COVID-19 epidemic were described by frequency and percentage for the categorical data, a mean with standard deviation (SD) for the parametric data, and a median with interquartile range (IQR) for the non-parametric data. A full exploratory analysis using a multivariable logistic regression was performed to determine factors associated with consistent performance of the measures and recommendations for COVID-19 prevention and control in schools. The results of this study were reported in accordance with the strengthening of the reporting of observational studies in Epidemiology (STROBE) checklist. All statistical analyses were two-sided, and a *p*-value ≤ 0.05 was considered statistically significant.

### 2.4. Ethical Considerations

This study was conducted in accordance with the Declaration of Helsinki guidelines and the protocol was approved by the Research Ethics Committee, Faculty of Medicine, Chiang Mai University, Thailand (Study Code: COM-2564-08506).

## 3. Results

### 3.1. Characteristics of the Participants

Out of the 402 study participants, the mean (SD) age of the participants was 38.4 (11.1) years of age. The majority of participants were female (74.4%). Most participants were affiliated with a private school, 78.9%, followed by a government school (16.2%), with the major role of the participant being described as teaching (90.1%: 362/402). Half of the participants (52.0%) had an income of less than 20,000 baht per month, and 122 (30.4%) reported having insufficient income (data not shown). The majority had an expense allowance for COVID-19 prevention of between100 to 300 baht per month (43.8%) and more than 300 baht per month (41.8%). Participants received complete vaccination against COVID-19 at a rate of 90.8% (365/402). The details of these characteristics are shown in Table 1.

### 3.2. Opinion Regarding the Measures and Recommendations for School Re-Opening

The majority of participants agreed to every measure and recommendation for the prevention of COVID-19 and the need for controls in schools (proportion range between 97.5 to 100%), and that they had consistently performed the necessary actions for more than 90% of the time. The measure that had the highest percentage as regards consistent performance was “You must strictly follow preventive measures such as washing your hands frequently, wearing a mask, and keeping distance between people. Avoid going to crowded places.” (97.8%). In contrast, the measure that had the lowest percentage of participant’s performed consistently was “You must communicate your knowledge about stress and the stress management process for students and personnel in educational institutions.” (91.3%) (Table 2).

Table 3 shows the perceptions and concerns of school employees regarding on-site education during the COVID-19 pandemic. Most participants agreed or strongly agreed that their colleagues and schools were ready for on-site education, with 91.3% and 94.5%, respectively. Concerns of participants regarding school opening recorded more than half of participants were very concerned with school opening but agree with school opening for 67.2%, followed by moderately concerned in 24.4% of cases, however, 22.4% of participants were very concerned and did not agree with school opening. From the school readiness interview, there were only 33 (45.8%) schools that passed the green and yellow criterion according to the recommended guidelines for the prevention of the spread of infectious diseases and were ready for re-opening (Figure 2).

### 3.3. Factors Affecting the Action of Participants

As shown in Table 4, a multiple logistic regression showed factors which significantly associated with the participant’s consistently performing the measures and recommendations for COVID-19 prevention and control in schools, included male gender (aOR 0.43, 95% CI 0.23 to 0.80; *p* = 0.008), high level of concern and disagreement with school re-opening (aOR 3.78, 95% CI 1.04 to 13.70; *p* = 0.043). At least one (aOR 0.04, 95% CI 0.01 to 0.19; *p* < 0.001) and two or more (aOR 0.03, 95% CI 0.01 to 0.023; *p* < 0.001) disagreements with the COVID-19 preventive and control measures and recommendations in schools were also substantially connected to inconsistent performance of all of the COVID-19 prevention and control measures and recommendations in schools.

## 4. Discussion

Individual agreements and actions on preventive measures can potentially indicate the ability of the community to cope with a pandemic situation [8,9,10]. In the context of an academic setting, the awareness and agreement of preventive measures for COVID-19 prevention and control must be consistent across students, caregivers, and those responsible for the supervision of students. All of the primary measures controlling entry and exit from schools, as well as the secondary measures regarding screening for abnormal symptoms, require the cooperation of a significant number of school employees to be successful in preventing the consequences of the COVID-19 outbreak in the school [16]. The lowest percentage of agreement on the measure was “You must perform a health screening for everyone who enters the school according to the procedures.” (97.5%), while the lowest proportion of consistent performance measure was “You must communicate your knowledge of stress, the stress management process for students and personnel in educational institutions” (91.3%). Although the percentages of agreement on preventive measures were relatively high among school employees, consistent performance on preventive measures is still not entirely achieved. The strongest factor related to inconsistent performance of the preventive measure to COVID-19 was disagreement with the measures. The more the disagreement with the measures, the greater the change in performing preventive measures inconsistently.

According to the context of Thailand, the social norm against risky behaviors or refusing to comply with preventive measures in public areas is commonly unacceptable, resulting in social pressure and legal penalties due to public health emergency regulations. It could explain the relatively high percentages of agreement and compliance among school employees in our study (more than 97.5%). This finding was also consistent with the study in Vietnam [8]. Focusing on the detail of the measure with the lowest percentage of agreement, it might state an unclear role of whom will be the main practitioner. The health screening for everyone who enters the school might need more than one person to operate effectively. Hesitation of responsibility in operation may influence agreement to the selected measure. The COVID-19 pandemic would affect both teacher’s and student’s psychosocial condition potentially causing stress, burn out, decreased of quality of life, and other damage to mental health [17,18]. Support regarding the management of changes in mental status is necessary [19,20]. As a result, inconsistent application of stress knowledge and management may occur as a result of a lack of details on how to communicate with a student and what are appropriate contents for a student in the various levels. Statements pertinent to these measures must be clearly defined in the details for practitioners, then the chances of performing the preventive measures consistently might be increased.

Gender was also found to be related to preventive behavior. Male participants tended to perform preventive measure less consistently than female participants, as well as the participants’ concern and disagreement with the re-opening of school was a positive factor for consistent performance on the preventive measures. The consistent performance of these participants could be attributed to their higher awareness of the pandemic situation than those who were slightly concerned. Health concern for others might encourage better adherence to some of preventive measure, and women tending to express more related concern to COVID-19 than men [21]. This might explain result of higher chance among women to consistently perform the preventive measures. Other factor that might related to the school employee’s preventive performance was the readiness to school re-opening, however, it showed non-statistically significance. The majority of participants reported that either the school or the school employees were ready for on-site education, however, the proportion of schools that able to follow the recommended guidelines for the prevention of the spread of infectious diseases and were ready for re-opening was only 45.8% (green and yellow criterion). This inconsistency between self-opinion and reality of readiness to school re-opening might explain that making decision for school re-opening need more evidences beyond only how were school employees perform preventive measures. All of these factors should be considered by academic institutions in order to accomplish the preventive measures more consistently. A lack of support for the practitioner to perform the preventive measures, even if the teachers agreed to the measures, could result in inconsistent performance, increase the risk of a new outbreak after the school reopened, and other consequences [22,23,24].

From the author’s knowledge, this study was firstly studied related factors to consistent performance of the preventive measures among school employees during the COVID-19 pandemic. There were several limitations to this study. First, according to the limitations of a cross-sectional study, the direction of associations cannot be determined, and the association between consistent performance of the measures and school employees’ concerns and disagreements should be carefully interpreted. Second, the percentage of agreements and actions on preventive measures may be overestimated due to self-reported questions and representative biases. Third, although the schools were limited to the urban district of Chiang Mai, the variety of school types and sizes in our study could represent the current situation in most schools in Thailand, particularly those in urban and suburban areas. Since the data was specific to the Thai setting at a given time of the pandemic, generalization of the results to any other context is not a matter of course and should be considered with caution. The study data were collected from November to December 2021, when the delta-strain-COVID-19 was decreasing in Thailand and vaccinations were more widely available, as evidenced by the 90.8% immunization rate among participants. The re-opening of schools to an on-site pattern in Chiang Mai was permitted, but was shortly canceled due to new clusters in schools following the arrival of Omicron-strain-COVID-19 [25]. Our study findings could inform that school employees need support from relevant sectors to strengthen compliance with preventive measures, which could result in the high effectiveness of COVID-19 prevention in schools and enhance the safety of school re-opening. Even among school employees, the percentage of preventative performance is high. The recently transmissible of the Omicron variant is the crucial risk for school re-opening [26]. As indicated by a multicountry review and a modeling study, the impacts of school reopening on disease transmission are strongly dependent on the prevalence of the disease in the surrounding community [16,27,28]. Other factors related to students and their parents, need to be evaluated further in order to develop effective disease-prevention strategies in schools.

## 5. Conclusions

The majority of participants agreed with the proposed COVID-19 preventive measures in schools. To strengthen compliance with preventive measures, school employees, particularly male personnel, should be supported by the relevant sectors to decrease disagreement with the measures and enhance awareness of pandemic situation before the school re-opening.

## Figures and Tables

**Figure 1 ijerph-19-05662-f001:**
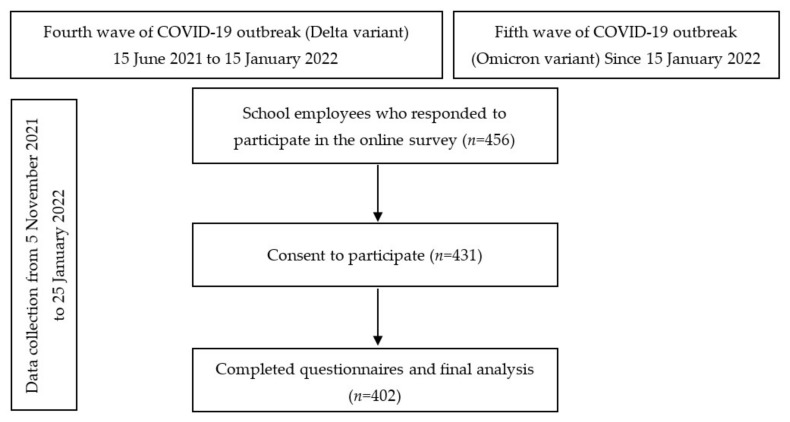
The study flow diagram and pandemic timeline.

**Figure 2 ijerph-19-05662-f002:**
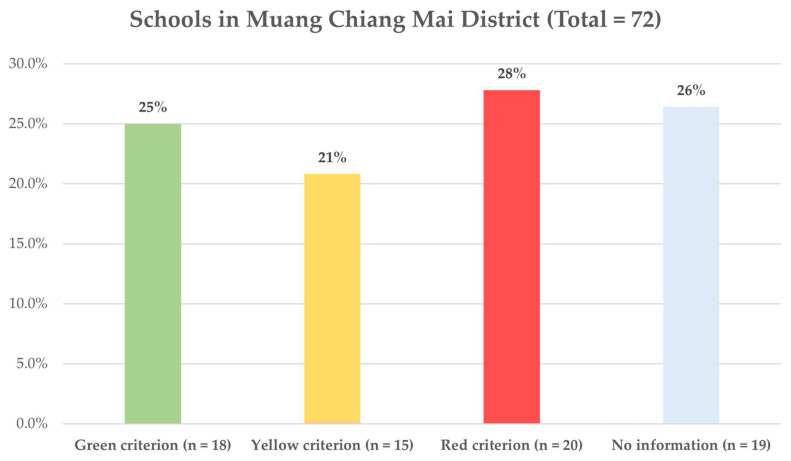
Proportion of schools in the urban district of Chiang Mai which meet the criteria recommended for the re-opening of on-site education. Green criterion, All primary or secondary dimensions are actionable; Yellow criterion, All primary dimensions and some of secondary dimensions are actionable; Red criterion, No primary or secondary dimensions actionable.

**Table 1 ijerph-19-05662-t001:** The characteristics of the participants.

Characteristics	*n* = 402	
	*n*	%
Age (years), mean ± SD	38.4	±11.1
Gender		
Female	299	74.4
Male	103	25.6
School affiliation		
Government school	65	16.2
Private school	317	78.9
Demonstration school	20	4.9
Length of working’s experience, median (IQR)	10	(4–20)
School roles		
Teaching	362	90.1
Non-teaching	40	9.9
Income (baht/month)		
≤20,000	209	52.0
20,001–30,000	119	29.6
30,001–40,000	28	7.0
40,001–50,000	30	7.4
50,001–60,000	12	3.0
>60,000	4	1.0
Cost for COVID-19 prevention (baht/month)		
<100	58	14.4
100–300	176	43.8
>300	168	41.8
COVID-19 vaccination		
No	11	2.7
Incomplete	26	6.5
Complete	365	90.8

**Table 2 ijerph-19-05662-t002:** Agreement and action regarding the measures and recommendations for the prevention and control of COVID-19 in school.

COVID-19 Prevention Measures	Agreement				Action					
	Agree		Disagree		Consistently Performed		Partially Performed		Not Performed	
	*n*	(%)	*n*	(%)	*n*	(%)	*n*	(%)	*n*	(%)
1. You must follow the information on the epidemic situation from reliable sources.	400	99.5	2	0.5	387	96.3	15	3.7	-	-
2. You must observe your own abnormal symptoms. If there are any respiratory symptoms, you should stop working and consult a medical team immediately.	400	99.5	2	0.5	388	96.5	11	2.7	3	0.8
3. You must strictly follow preventive measures such as washing your hands often, wearing a cloth mask or a mask and keeping distance between people and avoid going to crowded places.	402	100.0	-	-	393	97.8	9	2.2	-	-
4. You must inform parents of supervising students bringing their own personal items and protective equipment to school.	399	99.3	3	0.7	379	94.3	20	5.0	3	0.7
5. You must communicate knowledge. Advice or provision of public relations materials to prevent and reduce the risk of spreading COVID-19.	399	99.3	3	0.7	370	92.0	27	6.7	5	1.3
6. You must clean the teaching aids or equipment that is a high-risk touch point after every use.	397	98.8	5	1.2	379	94.3	19	4.7	4	1.0
7. You must supervise the seating arrangements within the school premises in accordance with the basic principle, the distance between people is at least 1–2 m.	399	99.3	3	0.7	382	95.0	18	4.5	2	0.5
8. You must inspect, supervise, and follow up on student attendance.	400	99.5	2	0.5	386	96.0	13	3.2	3	0.8
9. You must perform health screening for everyone who enters the school according to the procedures.	392	97.5	10	2.5	379	94.3	19	4.7	4	1.0
10. You must observe groups of students with behavioural problems or students who do not cooperate with the measures set by the teacher in order to receive assistance.	400	99.5	2	0.5	384	95.5	13	3.2	5	1.3
11. You must communicate your knowledge of stress and the stress management process for students and personnel in educational institutions.	396	98.5	6	1.5	367	91.3	30	7.5	5	1.2

**Table 3 ijerph-19-05662-t003:** Perceptions and concerns of school employees regarding on-site education during the COVID-19 pandemic.

**Perceptions of School’s Employees**	**Strongly Agree**		**Agree**		**Disagree**		**Strongly Disagree**	
	** *n* **	**(%)**	** *n* **	**(%)**	** *n* **	**(%)**	** *n* **	**(%)**
Your school was ready for on-site education.	186	46.3	181	45.0	24	6.0	11	2.7
Your school staff were ready for on-site education.	199	49.5	181	45.0	16	4.0	6	1.5
**School Employees Concerns**	**Very Concerned and not Agree with School Opening**		**Very Concerned but Agree with School Opening**		**Moderately Concerned and Agree with School Opening**		**Slightly Concerned**	
	** *n* **	**(%)**	** *n* **	**(%)**	** *n* **	**(%)**	** *n* **	**(%)**
Concerned regarding school re-opening for on-site education.	90	22.4	180	44.8	98	24.4	34	8.4

**Table 4 ijerph-19-05662-t004:** Factors affecting the participant’s consistently performing the measures and recommendations for COVID-19 prevention and control in schools.

Factors	aOR	95% CI	*p*-Value
Characteristics			
Age	1.03	0.98 to 1.08	0.263
Gender			
Male	0.43	0.23 to 0.80	0.008 **
Female	(ref.)		
School affiliation			
Private school	1.52	0.70 to 3.28	0.289
University demonstration school	0.76	0.16 to 3.42	0.724
Government school	(ref.)		
Length of working’s experience	0.98	0.93 to 1.04	0.558
School roles			
Teaching	1.39	0.55 to 3.52	0.487
Non-teaching	(ref.)		
Teachers’ perceptions and concern			
Your school was ready for on-site education			
Strongly agree	2.03	0.28 to 14.59	0.478
Agree	2.50	0.42 to 14.82	0.312
Disagree	1.17	0.18 to 7.35	0.870
Strongly disagree	(ref.)		
Your school staff were ready for on-site education			
Strongly agree	4.53	0.42 to 48.78	0.212
Agree	1.56	0.18 to 13.61	0.688
Disagree	1.85	0.18 to 19.17	0.606
Strongly disagree	(ref.)		
Concern regarding school opening for on-site education			
Very concerned and not in agreement with school opening	3.78	1.04 to 13.70	0.043 *
Very concerned but agree with school opening	1.79	0.60 to 5.33	0.298
Moderately concerned and agree with school opening	2.50	0.75 to 8.26	0.134
Slightly concerned	(ref.)		
Agreement regarding the measures and recommendations for the COVID-19 prevention and control in school			
≥2 disagreements	0.03	0.01 to 0.23	0.001 **
1 disagreement	0.04	0.01 to 0.19	<0.001 **
All agree	(ref.)		

aOR, adjusted Odds ratio obtained from multivariable logistic regression; CI, Confidence interval. Dependent variable is consistent performance of all of the measures and recommendations for the COVID-19 prevention and control in school (yes/no). * Significant association at *p* ≤ 0.05; ** Significant association at *p* < 0.001.

## Data Availability

The data presented in this study are available on request from the correspondent author.

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
