# Peer review of "Perspectives and Factors Affecting the Preventive Behavior Pertinent to COVID-19 among School Employees in Chiang Mai, Thailand: A Cross-Sectional Study"

_ijerph, 2022, doi:10.3390/ijerph19095662_

Round 1
Reviewer 1 Report
The paper presented is original with an interesting topic in the current reality.
The writing style is formal and appropriate. The introduction makes it possible to understand the theoretical bases of the study, although it could be improved with references from this same journal regarding the special issue Teachers Wellbeing and Covid already published.
The method is adequately described, although it would be convenient to add the psychometric properties of the 44-item questionnaire and if a validation process has been carried out.
The statements of the discussion are consistent with the results but there should be a greater emphasis of discussion with previous research. It is recommended to add more references in this section.
Author Response
Responses to Reviewer 1 Comments
We would like to thank the reviewer 1 for all the useful comments and suggestions. We have revised the manuscript based on these constructive suggestions. We believe that the revised manuscript is now greatly improved after the revisions have been made, as suggested by the reviewers and the editor.
Point 1: The paper presented is original with an interesting topic in the current reality.
Response 1: We would like to thank the reviewer for the positive comment.
Point 2: The writing style is formal and appropriate. The introduction makes it possible to understand the theoretical bases of the study, although it could be improved with references from this same journal regarding the special issue Teachers Wellbeing and Covid already published.
Response 2: We would like to thank the reviewer for the positive feedback and the suggestion. We have cited the relevant published studies belonging to the special issue Teachers Wellbeing and Covid (reference no.19).
References:
[19] Solís García, P.; Lago Urbano, R.; Real Castelao, S. Consequences of COVID-19 Confinement for Teachers: Family-Work Interactions, Technostress, and Perceived Organizational Support. Int. J. Environ. Res. Public Health 2021, 18, 11259.
Point 3: The method is adequately described, although it would be convenient to add the psychometric properties of the 44-item questionnaire and if a validation process has been carried out.
Response 3: We would like to thank the reviewer for the positive feedback and the suggestion. The questions were developed by the national committee, including health experts and educational experts.
We have provided the information of the questions for assessing school employees and school, and the reliability of these questionnaires in the section 2.2. Questionnaire design (Page 4, Line 159-165; Page 5, Line 171-173, Line 176-178, and Line 181-183) and the supplementary table S1 and S2.
Point 4: The statements of the discussion are consistent with the results but there should be a greater emphasis of discussion with previous research. It is recommended to add more references in this section.
Response 4: As the positive suggestion from the reviewer 1, we found one study from the same special issue which related to management on teachers’ mental health (reference no. 19). There are three studies relevance to the trajectories of outbreak after school reopened in various countries and its associated factors (reference no. 16,27,28).
References:
[16] Levinson, M.; Cevik, M.; Lipsitch, M. Reopening Primary Schools during the Pandemic. New England Journal of Medicine 2020, 383, 981-985, doi:10.1056/NEJMms2024920.
[27] Panovska-Griffiths, J.; Kerr, C.C.; Stuart, R.M.; Mistry, D.; Klein, D.J.; Viner, R.M.; Bonell, C. Determining the optimal strategy for reopening schools, the impact of test and trace interventions, and the risk of occurrence of a second COVID-19 epidemic wave in the UK: a modelling study. The Lancet Child & Adolescent Health 2020, 4, 817-827, doi:https://doi.org/10.1016/S2352-4642(20)30250-9.
[28] Goldhaber-Fiebert, J.D.; Studdert, D.M.; Mello, M.M. School Reopenings and the Community During the COVID-19 Pandemic. JAMA Health Forum 2020, 1, e201294-e201294, doi:10.1001/jamahealthforum.2020.1294.

Reviewer 2 Report
Abstract
The abstract gives a good overview of the research. Need to include the purpose, methods and results. Need to add the instrument which been used to assess the data and what the main output and implication. The objective of the research also need to be included in abstract.
Introduction
The problem statement of the research still not clear. The objective of research is not clear. Why this research contributes re-opening of schools.
Need to include the importance of this research from 82-89
any hypothesis for this research?
Materials and Methods
The total of population is not included. How this sample been taken. Is random or purposive sample. Any inclusive and exclusive in your sample.
The instrument of research is not explained. What is source of this instruments. How the reliability and validity of the instruments.
How the sample has been taken?
Results
It looks good.
Discussion
The is not been discussed with any theoretical approach for these studies. What is the base for this research?
Please include limitation of this research and what will be the main implication of this research.
Author Response
Responses to Reviewer 2 Comments
We would like to thank the reviewer 2 for all the useful comments and suggestions. We have revised the manuscript based upon these constructive suggestions. We believe that the revised manuscript is now greatly improved after the revisions have been made, as suggested by the reviewers and the editor.
Abstract
Point 1: The abstract gives a good overview of the research. Need to include the purpose, methods, and results. Need to add the instrument which been used to assess the data and what the main output and implication. The objective of the research also need to be included in abstract.
Response 1: We have added the information in the abstract as suggested by the reviewer 2’s comments.
Study objectives:
The study objectives are (1) to assess school employee agreement and actions in accordance with the preventive measures and recommendations for COVID-19 prevention and control, (2) to assess the readiness of the school and employees for on-site education, and (3) to determine factors associated with consistent performance of these measures. (Page 1, Line 16-20)
The method of data collection and instruments:
Self-reported online questionnaires were used to collect school employees’ data. The readiness of schools for on-site education was assessed using 44 indicators from the practical guidelines by the Thai Department of Health. (Page 1, Line 22-25)
Implications:
Our study findings can inform the stakeholders to decide on the re-opening and supportive strategies. (Page 1, Line 35-36)
Introduction
Point 2: The problem statement of the research still not clear. The objective of research is not clear. Why this research contributes re-opening of schools.
Response 2: We thank the reviewer 2 for pointing out unclear issues in the introduction. We have rephrased the last paragraph of the introduction to provide more concise explanation regarding the problems, study objectives, and how this study contributed to the re-opening of schools during the recent pandemic situation. It now reads “Due to a lack of information regarding agreement and compliance with preventive measures among key players, including school employees and schools, estimating the level of efficacy of the preventive measures and the risk of an outbreak following the school's reopening were challenging. This study aimed (1) to assess school employee agreement and actions in accordance with the preventive measures and recommendations for COVID-19 prevention and control, (2) to assess the readiness of the school and employees for on-site education, and (3) to determine factors associated with consistent performance of these measures. The overall situation during the COVID-19 pandemic is very changeable due to the rapid adaptation of SARS-CoV-2 variants [14]. The percentages of preventive measures performed by school employees and the level of school readiness could reflect the efficacy of the current preventive measures for school re-opening during the pandemic of COVID-19. Identifying factors related to school employees' compliance with preventive measures is also essential for enhancing their compliance. Our study findings can inform the stakeholders to decide on the re-opening and develop supportive strategies.” (Page 2-3, Line 97-111)
Point 3: Need to include the importance of this research from 82-89
Response 3: We have added the importance of this research in the last paragraph of the introduction as suggested by the reviewer 2’s comment. It now reads “Due to a lack of information regarding agreement and compliance with preventive measures among key players, including school employees and schools, estimating the level of efficacy of the preventive measures and the risk of an outbreak following the school's reopening were challenging.” (Page 2, Line 97-100)
Point 4: any hypothesis for this research?
Response 4: We hypothesized that the percentages of preventive measures performed by school employees and the level of school readiness could reflect the efficacy of the current preventive measures for school re-opening during the pandemic of COVID-19. Identifying factors related to school employees' compliance with preventive measures is also essential for enhancing their compliance.
Materials and Methods
Point 5: The total of population is not included. How this sample been taken. Is random or purposive sample. Any inclusive and exclusive in your sample.
Response 5: We would like to clarify that this study invited all school employees in all 72 schools located in urban district of district. They were informed to participate this online surveys if they had a responsibility for COVID-19 prevention in their schools. We have added the total of school employees in all 72 schools and the response rate in the section 2.1. Study Design and Participants and Figure 1. It now reads “This study invited 5767 school employees in all 72 schools located in urban district of Chiang Mai. They were informed to participate this online surveys via the school coordinators if they had a responsibility for COVID-19 prevention in their schools.” (Page 3, Line 128-131)
Point 6: The instrument of research is not explained. What is source of this instruments. How the reliability and validity of the instruments.
Response 6: The questions for assessing school employees' agreement and actions on COVID-19 preventive practices were designed in accordance with the measures and recommendations for school employees in the prevention of the spread of the epidemic of COVID-19 in Thailand issued by the department of health, the ministry of public health. Also, the questions for evaluating the readiness of schools for re-opening were designed following 44 indicators from the practical guidelines by the department of health and the ministry of education. These measures and recommendations were developed by the national committee, including public health experts and infectious disease specialists from the department of health, and educational experts from the ministry of education. The reliability test of 11 questions for assessing school employees’ agreements and actions in accordance with the preventive measures obtained a Cronbach’s coefficient alpha of 0.83 and 0.89, respectively. The reliability test of the six dimensions of school assessment obtained a Cronbach’s coefficient alpha between 0.76 and 0.82, respectively.
We have provided the aforementioned information and the reliability of these questionnaires in the section 2.2. Questionnaire design (Page 4, Line 159-165; Page 5, Line 171-173, Line 176-178, and Line 181-183) and the supplementary table S1 and S2.
Point 7: How the sample has been taken?
Response 7: We have provided the information of population and the derived participants on Page 3, Line 128-131.
Results
Point 8: It looks good.
Response 8: We would like to thank the reviewer for the positive feedback.
Discussion
Point 9: The is not been discussed with any theoretical approach for these studies. What is the base for this research?
Response 9: We regret if we misunderstood this point due to the absence of the subject of this phrase. We have additionally described about the theoretical knowledge for this study in the first paragraph of the discussion. It now reads “Individual agreements and actions on preventive measures can potentially indicate the ability of the community to cope with a pandemic situation [8-10].”(Page 9, Line 269-270)
Point 10: Please include limitation of this research and what will be the main implication of this research.
Response 10:
- We addressed the further limitation of this study in the discussion. It now reads “There were several limitations to this study. First, according to the limitations of a cross-sectional study, the direction of associations cannot be determined, and the association between consistent performance of the measures and school employees' concerns and disagreements should be carefully interpreted. Second, the percentage of agreements and actions on preventive measures may be overestimated due to self-reported questions and representative biases. Third, although the schools were limited to the urban district of Chiang Mai, the variety of school types and sizes in our study could represent the current situation in most schools in Thailand, particularly those in urban and suburban areas.” (Page 10, Line 329-337)
- We have added the implication of this study in the end of discussion. It now reads “Our study findings could inform that school employees need support from relevant sectors to strengthen compliance with preventive measures, which could result in the high effectiveness of COVID-19 prevention in schools and enhance the safety of school re-opening.” (Page 10, Line 342-345)
